# Preparation of *Dendrobium officinale* Flower Anthocyanin and Extended Lifespan in *Caenorhabditis elegans*

**DOI:** 10.3390/molecules27238608

**Published:** 2022-12-06

**Authors:** Shuangxi Li, Jianfeng Wang, Liangliang Zhang, Yang Zheng, Guorong Ma, Xiaoming Sun, Jianfeng Yuan

**Affiliations:** 1Xingzhi College, Zhejiang Normal University, Lanxi 321100, China; 2Zhejiang Lanxi Jinrong Biological Technology Co., Ltd., Lanxi 321100, China; 3Key Laboratory of Wildlife Biotechnology and Conservation and Utilization of Zhejiang Province, Zhejiang Normal University, Jinhua 321004, China

**Keywords:** *Dendrobium officinale*, anthocyanin, *Caenorhabditis elegans*, antioxidant, lifespan

## Abstract

The *Dendrobium officinale* flower is a non-medicinal part of the plant, rich in a variety of nutrients and bioactive ingredients. The purpose of this article was to explore the preparation conditions of anthocyanins (ACNs) from the *D. officinale* flower. Subsequently, its anti-aging effects were evaluated with *Caenorhabditis elegans*. Results showed that the ACNs had antioxidant activities on scavenging free radicals (DPPH· and ABTS^+^·), and the clearance rate was positively correlated with the dose. Additionally, ACNs significantly increased the activity of superoxide dismutase (SOD) in *C. elegans*, which was 2.068-fold higher than that of the control. Treatment with ACNs at 150 μL extended the lifespan of *C. elegans* by 56.25%, and treatment with ACNs at 50 μL promoted fecundity in *C. elegans*. Finally, the protective effect of ACNs enhanced stress resistance, thereby increasing the survival numbers of *C. elegans*, which provided insights for the development and practical application of functional products.

## 1. Introduction

Aging is an inevitable process experienced by living organisms which has attracted attention throughout the history of humankind [1]. Studies have shown that the aging process is a dynamic, chronological process characterized by the progressive accumulation of oxidative damage, leading to an imbalance in protein homeostasis and mitochondrial dysfunction [2]. Reactive oxygen and nitrogen species (RONS), such as superoxide anion (·O^2−^), singlet oxygen, hydroxyl radical (·OH), peroxide free radicals, and nitric oxide (NO), are produced by all aerobic cells. RONS, whether they are endogenous or exogenous, cause oxidative modification of the major cellular macromolecules (proteins, lipids, and DNA) [3]. The accumulation of RONS in the body results in the imbalance between the oxidation system and reduction system, and eventually leads to the aging and damage of cells [3,4].

Currently, *Caenorhabditis elegans* is a well-established model in genetics, which has been fully sequenced and is widely utilized in aging-related studies due to its short life cycle, rapid reproduction rate, and clear life extension mechanism [2,5,6]. It has been reported that some natural products have been shown to increase the lifespan of *C. elegans*, such as flavonoids [7,8,9,10], saponins [11,12], polysaccharides [13,14], cereal, and food-derived oil [15]. Anthocyanins (ACNs), a class of plant secondary metabolites, are natural pigments in plants which are stable in acid conditions [16]. In particular, ACNs contain reactive polyphenolic hydroxyl groups, which exhibit biological activities such as antioxidation, anti-inflammatory and anti-tumor activities, as well as reducing cardiovascular diseases, diabetes, and aiding in the prevention of obesity, all of which have regulatory effects on human health [16,17,18]. Therefore, ACNs have been widely used in food, health products, cosmetics, medicine, chemical industry, and other fields [19].

*Dendrobium* are sympodial epiphytic plants, belonging to the orchid genera. *Dendrobium* species have been used for a thousand years as first-rate herbs in traditional Chinese medicine (TCM). *Dendrobium officinale* Kimura et Migo, commonly known as “Tiepi shihu”, is an edible medicinal plant and has been used to prepare beverages, porridge, and soup for centuries [20]. A larger number of medical activities have been assigned to *D. officinale*, such as anti-inflammatory, anti-fibrotic, antioxidant, anti-diabetic, immunomodulatory, and anti-cancer activities [21,22,23]. In conjunction with the increasing production of *D. officinale* (tissue culture and greenhouse cultivation), there is a large amount of production of the flowers annually. The flowers of *D. officinale* have been traditionally used for tea making, cooking, or preparing medicinal liquor, which has not been fully utilized, resulting in a lot of waste of resources [24]. Studies have shown that *D. officinale* flower has a variety of physiologically active ingredients, such as polyphenols and volatile components; however, the usage of *D. officinale* flowers in healthcare-related products was limited until 2018, when Chinese authorities approved the *D. officinale* flower as a new food ingredient [20,25]. In the future, the introduction of new products using the *D. officinale* flower as a raw material is expected to expand.

In this study, we aimed to investigate whether ACNs had effects on anti-aging. Hence, we have firstly explored the extraction and purification process of ACNs, then studied the effects on extending the lifespan of *C. elegans*, which related to antioxidant effects, anti-stress abilities, and egg-laying and reproductive capacities. The results will provide a basis for the development and utilization of *D. officinale* flowers.

## 2. Results

### 2.1. ACNs Extraction and Purification

The extraction conditions for ACNs from *D. officinale* are shown in Figure 1. In this process, the pH (1.0, 2.0, 3.0, 4.0), solvent concentration (70%, 75%, 80%, 85%), material fluid ratio (1:10, 1:20, 1:30, 1:40), extraction temperature (40 °C, 50 °C, 60 °C, 70 °C), and extraction time (90 min, 120 min, 150 min, 180 min) were examined. Results showed that the optimal conditions were as follows: the pH value was 1.0, the ethanol concentration was 75%, the material fluid ratio was 1:20, the extraction temperature was 60 °C, and the extraction time was 120 min. Under this optimized condition, we obtained crude ACN extraction with bright red (Figure 1f), and the maximum absorption value of ACNs at 530 nm was 0.281, approximately 0.8324 mg/mL.

It has been reported that macroporous resin has a better purification effect on anthocyanins [26,27]. To obtain the purified *D. officinale* ACNs, seven different macroporous resins were selected for static adsorption and determined its adsorption capacity. The results are presented in Figure 2A. It could be observed that AB-8 had the highest adsorption capacity, reaching 5.5 mL/mL resin against other adsorbents. Hence, the further experiments were carried out using AB-8 resin as the absorbent. After crude ACNs were absorbed by AB-8, it was first washed with distilled water, then eluted with 80% ethanol, and the collected eluate was detected at 530 nm. Accordingly, the elution curve is analyzed in Figure 2. The elution peak of ACNs (main fraction) was presented in the collected tubes 10–15, which were collected and used for following experiments.

### 2.2. ACNs Antioxidant Activity In Vitro

The DPPH· and ABTS^+^·scavenging activities of ACNs is shown in Figure 3. Generally, the clearance of DPPH· radical was positively correlated with the amount of ACNs. When the addition amount of ACNs was 180 μL (approximately 0.3 mg/mL), the clearance effect on DPPH radical was the best, up to 96.78 ± 2.14%, while the DPPH· clearance of 60 μL (approximately 0.1 mg/mL) was 81.98 ± 2.71% (Figure 3A). In our previous study [28], the clearance effect of 0.1 mg/mL *Nymphaea hybrid* alcohol extracts and vitamin C on DPPH· radical was 41.71 ± 1.42% and 55.15 ± 2.21%, respectively. These results show that the activity of ACNs in scavenging DPPH· free radicals was higher than that of vitamin C. For the scavenging of ABTS^+^· free radicals, the clearance increased with the increasing ACN dose (within 80 μL). When the added ACNs were 80 μL, the maximum clearance rate was 95.54 ± 1.86% (Figure 3B). Overall, these results showed that *D. officinale* ACNs had positive antioxidant effects in vitro.

### 2.3. Antioxidant Properties In Vivo

The antioxidant enzyme activities in vivo were assayed using *C. elegans* as a model organism [29]. The relative values of the antioxidant enzymes SOD, CAT, and GSH-Px in *C. elegans* are shown in Table 1. Compared with the control group, the antioxidant enzymes of *C. elegans* in the trial group were enhanced. In particular for the SOD enzyme, the activity was 1.3087 ± 0.4451 U/mg prot, which was 2.068-fold higher than control (*p* < 0.05). However, there was no significant difference on the CAT and GSH-Px enzyme activities against the control trial. It showed that the *D. officinale* ACNs mainly enhanced the SOD enzyme activity in *C. elegans*, and had little effect on the CAT and GSH-Px enzymes.

### 2.4. Effect of ACNs on Longevity and Spawning

Aging will lead to a weakening of an organism’s metabolic renewal and repair ability, and brings many adverse effects in life, such as slow movement, decreased immunity, etc. [30]. In this study, we examined the effects of different doses of *D. officinale* ACNs on the longevity and spawning capacity of *C. elegans*, and the results are shown in Figure 4. As shown in Figure 4A, the different doses of *D. officinale* ACNs (50, 75, 100, and 150 μL) could extend the lifespan of *C. elegans*. Compared with the average lifespan of control (16 day), *D. officinale* ACNs increased longevity approximately 43.75%, 6.25%, 50%, and 56.25% in the trial group, respectively. However, in the 300 μL test group, it was found that the lifespan of the nematodes dropped to 12 days, which was only 75% of the control. Strangely, in the 75 μL test group, the longevity of the nematodes was lower than other test groups, and it was difficult for us to explain this phenomenon. Exposed with different doses of ACNs, the egg-laying of nematodes was significantly different (Figure 4B). The administered dose of 50 μL promoted the reproductive capacity of *C. elegans*, while the other trials had the opposite results. It is reported that the reproductive capacity was related to the DAF-16/FoxO, which could be decreased by ACNs [15]. Compared with control, the spawning numbers in the 75, 100, 150, and 300 μL trials decreased by 24.9%, 28.7%, 46.6%, and 74.5%, respectively. It was also strange that in the 75 μL, the peak egg-laying period of nematodes was 1 day earlier than that of the other groups. Overall, high dose of ACNs (300 μL) damaged the reproductive ability and affected the longevity of nematodes.

### 2.5. Effect of ACNs on Stress Resistance

#### 2.5.1. Heat Stress Resistance

Increasing ambient temperature can have a range of uncomfortable effects on an organism. According to the literature [8,31], when the body is in a high-temperature environment, the metabolism of cell tissue is affected, causing heat stress, which in turn affects lifespan. The longevity and survival of *C. elegans* after treatment with different doses of *D. officinale* ACNs are shown in Figure 5. It can be seen that the survival number of *C. elegans* was increased by being treated with ACNs, but it was decreased with 300 μL of ACNs. This result was consistent with previous results, mainly due to high doses of ACNs (Section 2.4). The average survival time of heat stress in the control group was 10 h, while that in the trial groups (50, 75, 100, and 150 μL) was more than 10 h. In particular, the trial group with 100 μL had the longest average survival time approximately 15 h after heat treatment, which was increased 50% more than the control group, indicating that *D. officinale* ACNs had a protective effect on *C. elegans* under heat stress.

#### 2.5.2. Oxidative Stress Resistance

The antioxidative active factors upon exogenous stimulation of *C. elegans* was reported to reduce oxidative damage in vivo [5]. This motivated us to speculate whether *D. officinale* ACNs can also alleviate oxidative stress in *C. elegans*.

In this study, H_2_O_2_ was used to induce hydroxyl radicals (·OH) generation and build up oxidative stress. Treated with different doses of *D. officinale* ACNs (50, 75, 100, 150, and 300 μL), *C. elegans* then was exposed to 1% H_2_O_2_, and we determined the number of *C. elegans* survival. Results showed that the survival amount of *C. elegans* was effectively increased with the treatment of 50 μL during the initial stage of oxidative stress (0–6 h), while other trials were in the opposite manner. As the culture progressed (exceed 10 h), the survival number of *C. elegans* was higher than control, except for the 300 μL trial group, which increased the death of *C. elegans* (Figure 6).

## 3. Discussion

Currently, the problem of an aging society is becoming more and more acute, accompanied with kinds of diseases related to aging [13]. Therefore, people try to find the best natural ingredient to treat these diseases and to improve quality of life. There is a growing interest in herbal remedies because of their effectiveness, minimal side effects, and relatively low cost [32,33]. *D. officinale* is a traditional Chinese herbal medicine, and its flower has been reported to be a source containing a large number of phenolic components [20,34], which are of great value in serving potential food functions. Pharmaceutical study revealed that the *D. officinale* flower exhibited antioxidant capacities and antihypertensive effects [35]. ACN, a water-soluble pigment, was first used in the food industry as a natural colorant. Actually belonging to a subclass of polyphenols, ACN is a 3,5,7-trihydroxy-2-phenylbenzopyran-type cation and exists as glycosylated and acylated anthocyanin aglycones. The presented C_6_-C_3_-C_6_ structural feature well explains the antioxidant properties and notable healthy effects of ACNs [16].

In this study, the ACNs from the *D. officinale* flower were prepared by solvent extraction coupled with adsorption purification. ACN is a polar compound; thus, the solvents used for the extraction are acidified aqueous solvents [36] or mixed with methanol, ethanol, or acetone [26,27,37]. Herein, we explored solid–liquid extraction process conditions and the results showed that the 1 part of material (flower powder) was extracted with 20 parts of solvent (75% ethanol solution) at 60 °C, pH 1.0 for 120 min, and 0.8325 mg/mL ACN was obtained. Studies showed that acidic pH prevents the degradation of the ACN pigments. Thus, hydrochloric [26,38], formic, or acetic acids [39] could be added to the extraction solvent. Compared to the literature, we employed citric acid during the extraction process, which not only could effectively adjust the pH, but had a milder acidity [40]. In addition, there are also ultrasonic-assisted extraction, microwave-assisted extraction, liquid static high-pressure-assisted extraction, aqueous two-phase extraction, and supercritical fluid extraction techniques [41]. Each of them has advantages and disadvantages.

There are several techniques used for the purification of ACNs, such as chromatography [42], membrane filtration [43], and adsorption [26,27]. In this paper, the obtained crude extract of ACN was subsequently purified using macroporous resins. Seven different macroporous resins were selected and their adsorption capacity determined, in which AB-8 showed the highest adsorption capacity, reaching 5.5 mL/mL resin. This can be attributed to its weak polarity, similar to ACNs, and to its high specific surface area (480~520 m^2^/g), which has a special selectivity for ACNs. After being eluted with 80% ethanol on an AB-8 column, the main ACNs were gained; however, the main fraction should be analyzed by HPLC and mass spectrometry to identify the collected compound.

In order to verify the antioxidant activity of ACNs from *D. officinale*, in vitro and in vivo experiments were carried out. We showed that ACNs could act as the free radical scavenger to reduce the DPPH· and ABTS^+^· in a dose-dependent manner. These results are consistent with previous findings of ACNs from other natural plants [16,20,36]. The DPPH· and ABTS^+^· radical scavenging has been proven to be mediated by hydrogen atom transfer (HAT) [44,45] and electron transfer (ET) [46], respectively. In fact, the assay of DPPH· and ABTS^+^· scavenging activities has been successfully applied to evaluate the antioxidant capacities of flavonoids and ACNs in vitro [47]. Therefore, the antioxidant enzyme activities of *C. elegans* were measured in vivo (Table 1). We observed that the antioxidant enzyme activities were increased with ACNs. Notably, the SOD activity reached 1.3087 ± 0.4451 U/mg prot (*p* < 0.05). SOD and CAT, two major antioxidant enzymes in *C. elegans*, have the main function of scavenging superoxide free radicals and H_2_O_2_ [5]. The results indicated that the ACNs from *D. officinale* mainly increased the SOD activity, while the effect on CAT and GSH-Px activity was not obvious.

In a previous study, ACNs were shown to reduce the amount of ROS, and to render the worms more tolerant against heat and oxidative stress [12,45]. We observed that ACNs from *D. officinale* could relieve stress, increasing the average survival time under heat stress by approximately 50%, and efficiently resisting the oxidative stress induced by H_2_O_2_ within 6 h. It has been reported that prolonging lifespan and mediating aging is facilitated by SKN-1 [5,8] or by the insulin/IGF-like pathway that regulates the DAF-16 factor [9,30]. However, in this study, the mechanism of antioxidation and anti-aging effects has not been thoroughly studied, which will motivate us to carry out further research.

## 4. Materials and Methods

### 4.1. Materials

The *D. officinale* flowers were harvested in May 2022 from Lanxi Jinrong Biological Technology Co., Ltd. Zhejiang, China. SOD, CAT, and GSH-Px assay kits were purchased from Nanjing Jian-Cheng Institute of Biological Engineering. All chemicals were of analytical grade and purchased from Sangon Biotech (Shanghai) Co., Ltd., Shanghai, China.

### 4.2. Extraction of the ACNs

The fresh *D. officinale* flowers were freeze-dried and crushed at low temperature. The ACNs were separated by the solvent extraction method (SEM) and the extraction parameters, such as pH, solvent concentration, material fluid ratio, temperature, and time, were investigated. According to the literature [27], the *D. officinale* flowers sample powder was extracted with the 75% ethanol aqueous solution at 60 °C for 120 min. Then, the solution was cooled down at room temperature and centrifuged at 8000 rpm for 15 min. The supernatant was collected and the sediment was repeatedly extracted following the above procedures. Finally, the crude extract of ACN was combined and stored at 4 °C. ACN concentration was determined according to the pH differential method [27], which was calculated using the following equation:(1)$$\mathrm{ACNs}\;\mathrm{concentration}\;(\mathrm{mg/L})=\frac{A\times M_{w}\times DF\times 1000}{\varepsilon \times L}$$
where *A*: (*A*_530 nm_−*A*_700 nm_)_pH1.0_-(*A*_530 nm_−*A*_700 nm_)_pH4.5_; *M_w_*: 449.2 g/mol (standard molecular weight of cyanidin 3-glucoside); *DF*: dilution factor; conversion coefficient from 1000: g to mg; *ε*: molar extinction coefficient, 26,900 L/cm/mol; *L*: cuvette light path length, cm.

### 4.3. Purification of the ACNs

The ACN purification process was carried out by adsorption method. First, to select the proper resin, 7 different resins (HDP-100, HDP-450, HDP-600, X-5, AB-8, D-101, and NKA-9) tested the static adsorption in the following manner [26]. All adsorbents were performed by treatment with 2 bed volumes (BV) of distilled ethanol overnight followed by rinsing with 4 BV of distilled water. An amount of 2.0 mL of each activated adsorbent was contacted with 20 mL of crude ACN in 250 mL flask on a rotating shaker for 2 h. After adsorption, the remaining ACNs in solution were determined by spectrophotometry (UVmini-1240, SHIMADZU, Japan). The adsorption capacity of each resin was calculated to screen the porous resin.

Dynamic adsorption was performed as follows [47]: 30 mL of selected adsorbent (AB-8) was packed in a glass column (20 mm × 250 mm). Hence, the BV was 30 mL and packed column length of the adsorbent was 16.8 cm. The crude ACNs were passed through the packed column at 3.0 mL/min until the adsorption reached saturation. Then, that was washed with distilled water (5 BV) followed by elution using aqueous ethanol (80%, *v/v*) at the flow rate of 1.0 mL/min. To obtain purified ACNs, fractions of 5.0 mL were collected and the absorption value determined at 530 nm.

### 4.4. In Vitro Antioxidation Assay

The DPPH· free radical clearance activity was determined according to our previous work [5]. The 0.3 mL sample and 2.7 mL DPPH solution (dissolved in methanol, 0.5 mM) were mixed, letting it stand and avoiding light reaction for 20 min. Taking blank as negative control, the absorbance values at 519 nm were measured. The clearance rate was calculated as follows: where *A*_0_, *A*_s_ are the absorbance value of blank and samples.
(2)$$\mathrm{DPPH}\cdot \;\mathrm{clearance}\;\mathrm{rate}(\%)=\frac{A_{0}-A_{s}}{A_{0}}\times 100$$

7.4 mM of ABTS diammonium and 2.6 mM of potassium perbisulfite (K_2_S_2_O_8_) were mixed, standing for 16 h at room temperature in darkness. The mixture was again diluted with phosphate buffer (pH 7.4) to the absorbance value at 734 nm of 0.700 ± 0.002, preparing the ABTS^+^· working solution, and then 0.1 mL of sample and 3.9 mL of ABTS^+^· solution were mixed and the absorbance was measured at 734 nm after reaction for 6 min, where *A*_0_, *A*_s_ are the absorbance value of blank and samples.
(3)$${\mathrm{ABTS}}^{+}\cdot \;\mathrm{clearance}\;\mathrm{rate}(\%)=\frac{A_{0}-A_{s}}{A_{0}}\times 100$$

### 4.5. Effect of ACNs on Antioxidant Enzyme Activity in C. elegans

The *C. elegans* N2 was routinely maintained at 20 °C on nematode growth medium (NGM) seeded with *E. coli* OP50 as nutrient [5]. Age synchronized nematodes were achieved by treating gravid hermaphrodites with bleach (10% sodium hypochlorite: 1 M sodium hydroxide = 1:1). The eggs were collected and incubated in buffer M9 for one day until the embryos hatched at the L1 stage, and then they were transferred to new fresh NGM plates until L4 synchronized nematodes were obtained.

To assay the antioxidant enzyme activity in vivo, 0.15 mL of ACNs were spread on L2 age-synchronized worms and gained L4-aged worms until cultured at 20 °C for 2 d, while M9 buffer was used as control. Subsequently, GSH-Px, SOD, and CAT in *C. elegans* were assayed according to the operation requirements of the kits (Nanjing Jian-Cheng Institute of Biological Engineering, Nanjing, China).

### 4.6. Lifespan Assay

The synchronized L2-aged worms were randomly divided into 6 groups, each with 3 plates including the control and test group. Thirty worms per plate were administered at the ACNs as 0, 50, 75, 100, 150, and 300 μL and the first day was considered day one. Thereafter, survival was examined daily by first touching posterior and then anterior. Worms that were scored as alive were transferred into a fresh incubation medium plate. The worm death standard was that platinum shovels touched worms without any response and worms displaying internal hatching or protruding organs were excluded.

### 4.7. Fecundity Assay

Age-synchronized L2 worms were treated as per the Section 4.6 method and two worms were randomly picked out from each group, then cultured separately. Worms were moved to new plates every day until the end of the reproductive period and the number of progeny in the original plates were counted after 24 h to allow all the fertile eggs to hatch [48].

### 4.8. Assessment of Stress Resistance

After being treated as per the Section 4.6 method, the age-synchronized L2 worms were cultured at 20 °C for 2 d. Subsequently, *C. elegans* worms were treated with 1% H_2_O_2_ or against evaporation (37 °C) to induce oxidative and thermal stress, respectively. Survival was counted hourly until all the worms died, and the trial with no ACNs treated was used as control group.

### 4.9. Statistical Analysis

Statistical analysis was carried out using SPSS ver. 20.0 (SPSS Inc., IBM Corporation, Chicago, IL, USA), and the comparisons of differences between the means of the treatments were tested by one-way analysis of variance (ANOVA) at a significance level of *p* < 0.05.

## 5. Conclusions

In this study, we first prepared ACNs from the *D. officinale* flower. Successful results of parameters of ACN extraction and purification were obtained. Then, the ACNs displayed a certain ability to scavenge free radicals in vitro, such as DPPH· and ABTS^+^·. Concurrently, ACNs also showed good potential to enhance the activity of antioxidant enzymes, especially the SOD activity in *C. elegans*. More importantly, *C. elegans* nematodes treated with ACNs showed the capacity to resist oxidative and thermal stress and displayed enhanced fecundity, resulting in the extended longevity of *C. elegans*. However, the details of ACN composition should be further analyzed on HRLC/MS or GC/LCMS, and the cytotoxicity on normal cells should also be detected to confirm the safety of ACNs. In conclusion, we have shown that ACNs have an antioxidant effect in vitro and in vivo, which can prolong the life of *C. elegans* and alleviate oxidative or thermal stress. This shows that ACNs have potential for antioxidant and anti-aging activities, which can be made as functional ingredients and enhance the value of *D. officinale*.

## Figures and Tables

**Figure 1 molecules-27-08608-f001:**
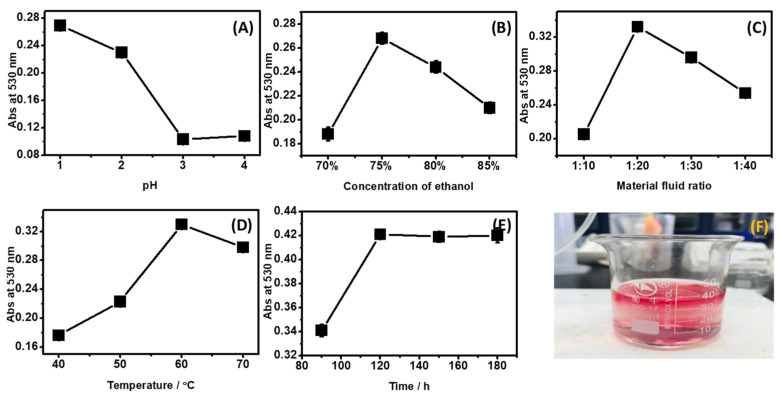
Extraction conditions for ACNs from *D. officinale*. (**A**) pH, (**B**) concentration of ethanol, (**C**) material fluid ratio, (**D**) temperature, (**E**) extraction time, (**F**) ACNs crude extracts.

**Figure 2 molecules-27-08608-f002:**
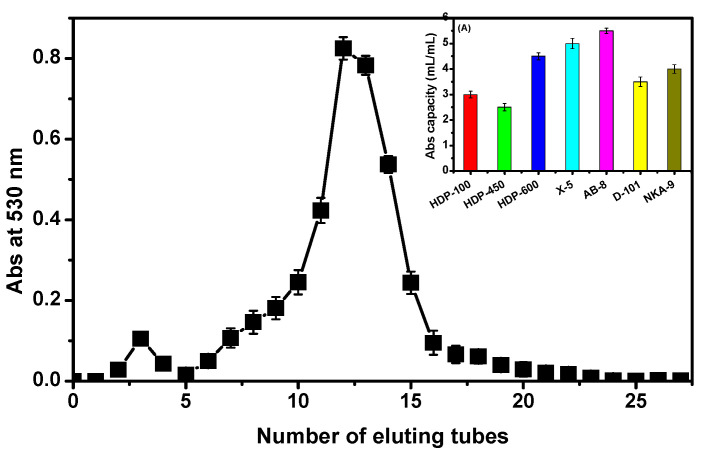
Elution curve of *D. officinale* ACNs by AB-8 resins. (**A**) Adsorption capacities of *D. officinale* ACNs on 7 different macroporous resins.

**Figure 3 molecules-27-08608-f003:**
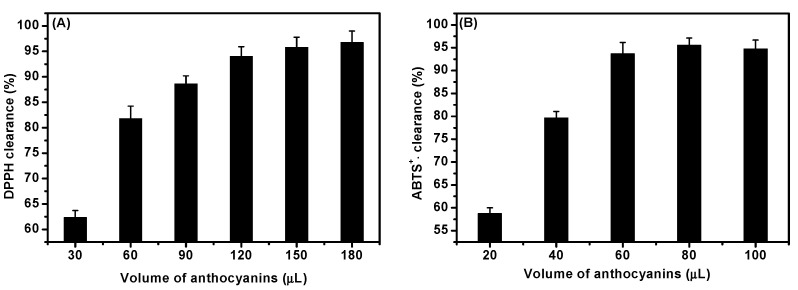
The free radical clearance effect on (**A**) DPPH· and (**B**) ABTS^+^· with different amounts of *D. officinale* ACNs.

**Figure 4 molecules-27-08608-f004:**
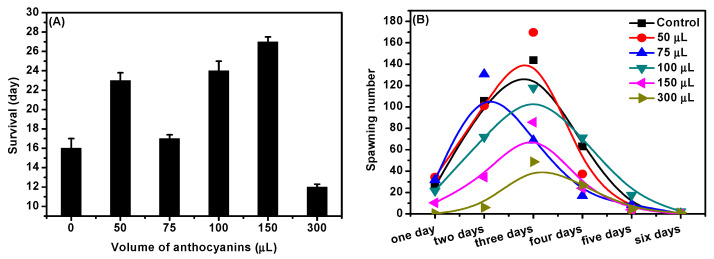
Effect of *D. officinale* ACNs on (**A**) longevity and (**B**) the spawning number in *C. elegans*.

**Figure 5 molecules-27-08608-f005:**
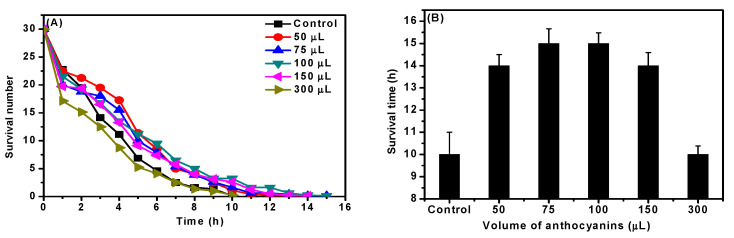
Effect of *D. officinale* ACNs on (**A**) heat stress and (**B**) the survival time in *C. elegans*.

**Figure 6 molecules-27-08608-f006:**
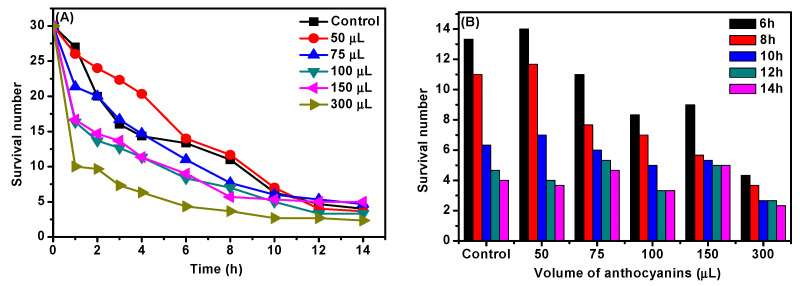
Effect of *D. officinale* ACNs on (**A**) oxidative stress and (**B**) the survival number at different time in *C. elegans*.

**Table 1 molecules-27-08608-t001:** Effect of *D*. *officinale* ACNs on antioxidant enzyme activity in *C. elegans*.

	SOD Activity (U/mg Prot)	CAT Activity (U/mg Prot)	GSH-Px Activity (U/mg Prot)
*D. officinale* ACNs	1.3087 ± 0.4451 ^a^	0.0578 ± 0.0091 ^b^	0.5641 ± 0.0415 ^b^
Control	0.6129 ± 0.0828	0.0551 ± 0.0052	0.5385 ± 0.0381

Note: Different letters on the same line indicate significant differences (^a^ *p* < 0.05; ^b^ No significant difference).

## Data Availability

Data reported in this study are contained within the article. The underlying raw data are available on request from the corresponding author.

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
