# Peer review of "Preparation of Dendrobium officinale Flower Anthocyanin and Extended Lifespan in Caenorhabditis elegans"

_molecules, 2022, doi:10.3390/molecules27238608_

Round 1

Reviewer 1 Report

The manuscript entitled "Preparation of Dendrobium officinale flower anthocyanin and extended lifespan in Caenorhabditis elegans" is of significance as it tries to explain the causes of aging and measures to delay it. However, there are some corrections needed in the manuscript.

1. The mechanism by which reactive oxygen and nitrogen species induce aging should be mentioned in the introduction as the study focus on it.

2. Add a review of the pharmacological effects of anthocyanin in the introduction

3. Taxonomic position of the plant should be mentioned in the introduction

4. The reproductive capacity in C. elegans went down with increasing concentration of ACNs. Could be it due to any toxic effect as seen in the case with 300µl? How that possibility can be ruled out rather than using the term “strange”?

5. Some references should be cited from the original paper rather than a review article

Other minor suggestions are highlighted in the attached file.

Author Response

Point 1: The mechanism by which reactive oxygen and nitrogen species induce aging should be mentioned in the introduction as the study focus on it.

Response 1: Thanks giving the comments. To our knowledge, there is a complete set of oxidation system and reduction system in the cell. The oxidation system mainly includes two major "families", reactive oxygen species / reactive nitrogen species (ROS / RNS), and the reduction system includes macromolecular antioxidant enzymes and small-molecule antioxidants. Normally, the oxidation system is in balance with the reduction system to maintain normal cell function. Once it loses the balance, and the cellular redox environment is damaged, showing stress and functional abnormalities, and the continuous imbalance will lead to aging and related diseases. In the introduction section, we have mentioned the theory of the aging (marked in red), which are induced by RONS.

Point 2: Add a review of the pharmacological effects of anthocyanin in the introduction.

Response 2: Thanks for the comment. We have reviewed the pharmacological effects of ACNs, including antioxidation, anti-inflammatory, anti-tumor, reducing cardiovascular diseases, diabetes and the prevention of obesity in the introduction section (Page 1, Line 45).

Point 3: Taxonomic position of the plant should be mentioned in the introduction.

Response 3: Thanks a lot. To a certain extent, the taxonomic position of D. officinale plays an important role. We have revised and mentioned it in article (Page 2, Line 49-54).

Point 4: The reproductive capacity in C. elegans went down with increasing concentration of ACNs. Could be it due to any toxic effect as seen in the case with 300 µL? How that possibility can be ruled out rather than using the term “strange”?

Response 4: Thanks for the comment. In the test of the effect of ACNs on the spawning, we found that the high concentration of ACNs, such as 300 µL, could decrease the reproductive capacity of C. elegans. Since no cytotoxicity experiments were performed (shortcoming), we can not give the conclusion whether the high concentration of ACNs has the toxic effect to C. elegans. However, according to the reference, we found that the reproductive capacity of C. elegans was related to DAF-16/FoxO, while the high concentration of ACNs could decrease the level of DAF-16/FoxO.

The term “strange” was used in the experiment of longevity. When different concentration of ACNs was used, the longevity of C. elegans was increased, but in the trail of 75 μL, the longevity was lower than other groups, though that was higher than control (Page 4, Line 134-135).

Point 5: Some references should be cited from the original paper rather than a review article.

Response 5: Thanks a lot for the suggestion. We have revised and cited the original references in Page 2, Line 53.

Point 6: Other minor suggestions are highlighted in the attached file.

Response 6: Thanks for the comment, and we have revised the some mistakes in the article. We should be more careful during the MS preparation.

Reviewer 2 Report

The authors in this MS focused on the optimization of extraction method of anthocyanin from Dendrobium officinale flower and test their effect in Caenorhabditis elegans to extent their lifespan.

-        This reference should be added to your introduction. It documents the antioxidant activity of phenolic compounds in Caenorhabditis elegans

(Identification of phenolic secondary metabolites from Schotia brachypetala Sond. (Fabaceae) and demonstration of their antioxidant activities in Caenorhabditis elegans PeerJ, 2016 | Journal article, DOI: 10.7717/peerj.2404)

In methodology

-        Authors should clarify the extraction method. the mentioned by solvent extraction but they didn’t specify how? Is it simple soaking, percolation ,………?

-        Measuring absorption at 530 nm doesn’t confirm it is related to anthocyanin 100%?

-        The isolation method doesn’t confirm that it is pure mixture of anthocyanin?

-        It was much better if the isolated mixture of anthocyanin was analysed on HRLCMS to explore the exact content qualitatively and quantically of anthocyanin content.

-        In conclusion: don’t repeat the results and give only the conclusion.

-        In biological parts all experiments are well performed. However, it was much better to test the cytotoxicity on normal skin cell line to check the safety of investigated compounds on local treatment before to go to in vivo assessment.

-        In general, the article is of great importance to have antiaging anti-oxidant components from natural source especially from waste plant part.

Author Response

Point 1: This reference should be added to your introduction. It documents the antioxidant activity of phenolic compounds in Caenorhabditis elegans. (Identification of phenolic secondary metabolites from Schotia brachypetala Sond. (Fabaceae) and demonstration of their antioxidant activities in Caenorhabditis elegans. PeerJ, 2016 | Journal article, DOI: 10.7717/peerj.2404).

Response 1: Thanks. We have cited the reference in the article. This reference focused on the identification of phenolic compounds from Schotia brachypetala, which showed the antioxidant activity in C. elegans. In addition, there has been confirmed the composition of the phenolic secondary metabolites by the means of LC/HRESI/MS/MS, which is a reference for us to further verify the composition and structure of ACNs.

Point 2: Authors should clarify the extraction method. the mentioned by solvent extraction but they didn’t specify how? Is it simple soaking, percolation ,………?

Response 2: Thanks very much. In this article, we used the solvent extraction method (SEM) to extract the ACNs, which was described in section 4.2. This is a solvent soaking method. The D. officinale flowers sample powder was extracted with the 75% ethanol aqueous solution at 60 °C for 120 min. Then the solution was cooled down at room temperature and centrifuged at 8000 rpm for 15 min. The supernatant was collected and the sediment was repeatedly extracted following the above procedures. Finally, the crude extract of ACNs was combined and stored at 4 °C. 

Point 3: Measuring absorption at 530 nm doesn’t confirm it is related to anthocyanin 100%?

Response 3: Thanks for the comment. We have performed a full-band scan on ACNs, and the results showed that there is a maximum absorption at 530 nm. In this article, we used the absorption at 530 nm to evaluate the relative concentration of ACNs, and determined the ACNs according to the pH differential method at 530 nm and 700 nm.

Point 4: The isolation method doesn’t confirm that it is pure mixture of anthocyanin?

Response 4: Thanks giving us this question. We extract the ACNs according to the reference. This is a solvent soaking method. So it is difficult to confirm the pure mixture of ACNs. Though we have purified the ACNs, but maybe there is the mixture of the ACNs.

Point 5: It was much better if the isolated mixture of anthocyanin was analysed on HRLCMS to explore the exact content qualitatively and quantically of anthocyanin content.

Response 5: Thanks for giving us the most important comments, which will inspire us to carry out further research. In the next step, we will test on the ACNs content, including number of hydroxyl groups, number of sugar groups, and structure of sugar, which will help us better to understand the anti-aging mechanism of ACNs. Now we have revised the article and clarified the shortcomings in the conclusion section, as well as the research next to be carried out.

Point 6: In conclusion: don’t repeat the results and give only the conclusion.

Response 6: Thanks for the comment. Duplicating the results in the conclusions section appears to be too cumbersome, therefore, we have made a revision to delete the detail results.

Point 7: In biological parts all experiments are well performed. However, it was much better to test the cytotoxicity on normal skin cell line to check the safety of investigated compounds on local treatment before to go to in vivo assessment.

Response 7: Thanks for giving us the constructive comments. Cytotoxicity is an important issue in the development of natural functional ingredients. This is also to be confirmed after the structural identification of ACNs.

Point 8: In general, the article is of great importance to have antiaging anti-oxidant components from natural source especially from waste plant part.

Response 8: Thanks. The reason why we choose the flowers of Dendrobium candidum as raw material is that a large number of Dendrobium flowers are produced every year, and these flowers have not been fully developed and utilized.

Reviewer 3 Report

Dear Authors

The MS entitled “Preparation of Dendrobium officinale flower anthocyanin and extended lifespan in Caenorhabditis elegans” was thoroughly reviewed. The MS is technically correct and has been well formulated. Some major corrections however, should be made. My concerns/suggestions in the MS are provided.

1. correct the font size n=and style throughout the MS.

2. In the introduction part, Caenorhabditis elegans model should be briefly discussed and justified why the authors selected this model? Also, the anthocyanins chemical nature should be added.

3.Were the anthocyanins stable at pH 1? Did the authors perform stability studies? For how many days they were stable in such lower pH?

4. page 8, 4.8 section. Add the normal life span of C. elegans and their death days after oxidative stress.

5. Why 300 μL dose was not effective? Did the authors carried some experiments (GC/LCMS analysis) to find out the chemical composition of the anthocyanins? Was there any reference drug/material used in these models? It is always better to have found chemical composition of extracts.

6. What might be the antioxidant mechanism of these extracts? Need to be included in the discussion part.

7.  Correct the fonts sizes of equations.

8. Antioxidation assay in vitro should be in vitro Antioxidation assay

9. The flowers as a whole were used for extraction or the petals?

Author Response

Point 1: correct the font size n=and style throughout the MS.

Response 1: Thanks. We have revised it throughout the article.

Point 2: In the introduction part, Caenorhabditis elegans model should be briefly discussed and justified why the authors selected this model? Also, the anthocyanins chemical nature should be added.

Response 2: Thanks for the comment. We have condensed the description of C. elegans and explained why the nematode was chosen as the model. Several organisms are used for antiaging studies, such as yeasts, C. elegans, flies, killifish, mice and rats. The C. elegans has a 3 days generation cycle, and can be easily manipulated. In addition, the statistical genetics and genomics information can be obtained through extensive online tools. Therefore, it is possible to construct mutations in C. elegans to simulate human aging and aging-related diseases using the genetic engineering.

The chemical properties of ACNs is also added in the introduction part (Page 1, Line 42-43).

Point 3: Were the anthocyanins stable at pH 1? Did the authors perform stability studies? For how many days they were stable in such lower pH?

Response 3: Thanks very much. In preliminary experiments, we have determined the stability of ACNs at pH 1.0 (data not show). It can maintain the red-pink color at pH 1.0 at 4 °C for more than 3 months. But if the pH shift to above 4.0, it will be soon changed the color to yellow brown.

Point 4: page 8, 4.8 section. Add the normal life span of C. elegans and their death days after oxidative stress.

Response 4: Yes. Thanks very much, and we made the mistake. In section 4.8, the control group was not mentioned, but the result was showed. Now we have added a description of the control group.

Point 5: Why 300 μL dose was not effective? Did the authors carried some experiments (GC/LCMS analysis) to find out the chemical composition of the anthocyanins? Was there any reference drug/material used in these models? It is always better to have found chemical composition of extracts.

Response 5: Thanks for giving us the suggestion. We have not determined the chemical composition of the ACNs yet. So we can not make the conclusion whether the ACNs has any toxic effect in the case with 300 µL. Nevertheless, the reference show that high concentration of ACNs may inhibit the DAF-16 with sufficient food, which will affect the reproductive capacity of C. elegans.

Point 6: What might be the antioxidant mechanism of these extracts? Need to be included in the discussion part.

Response 6: Thanks very much. We have mentioned that the prolongation of lifespan and mediating aging is by SKN-1or by the insulin/IGF-like pathway that regulates the DAF-16 factor according to the reference. But in this article, we have not confirmed it yet.

Point 7: Correct the fonts sizes of equations.

Response 7: Thanks. We have checked “Instruction for authors” and “Molecules-template”, where the font and the font size of equation is “Palatino Linotype” and 10, respectively. We have modified the font size.

Point 8: Antioxidation assay in vitro should be in vitro Antioxidation assay

Response 8: Thanks for the suggestion, and we have modified the title of 4.4 section.

Point 9: The flowers as a whole were used for extraction or the petals?

Response 9: Thanks for the question. The whole D. officinale flowers, with calyx, ovary, receptacle and stalk were used, and then that was freeze-dried and crushed at low temperature.

Round 2

Reviewer 2 Report

Currently, the paper is generally well written and structured. However, the authors responded to the most important comments regarding the identification the components of the ACNs mixture or testing the safety of this mixture in vitro as a future work. Otherwise, the paper is highly improved. 

Author Response

Point 1: Currently, the paper is generally well written and structured. However, the authors responded to the most important comments regarding the identification the components of the ACNs mixture or testing the safety of this mixture in vitro as a future work. Otherwise, the paper is highly improved.

Response 1: Thanks for the constructive comment, and this has attracted our extra attention. Though the ACNs from Dendrobium officinale flower has shown the good anti-aging effect, the components of the ACNs have not been tested yet. We are well aware that for the development of functional food ingredients, the composition of the substance and its toxicity to cells are very critical. Therefore, we will conduct the further study in detail, not only limited to the composition and cytotoxicity of ACNs, but also the mechanism of anti-aging of the main components.
